# Effects of l-Carnitine Supplementation in Patients Receiving Hemodialysis or Peritoneal Dialysis

**DOI:** 10.3390/nu12113371

**Published:** 2020-11-01

**Authors:** Masako Kuwasawa-Iwasaki, Hiroaki Io, Masahiro Muto, Saki Ichikawa, Keiichi Wakabayashi, Reo Kanda, Junichiro Nakata, Nao Nohara, Yasuhiko Tomino, Yusuke Suzuki

**Affiliations:** 1Department of Nephrology, Juntendo University Faculty of Medicine, Tokyo 113-8412, Japan; mkuwaza@juntendo.ac.jp (M.K.-I.); mmutou@juntendo.ac.jp (M.M.); sk-ojima@juntendo.ac.jp (S.I.); kiwakaba@juntendo.ac.jp (K.W.); reoreo@juntendo.ac.jp (R.K.); jnakata@juntendo.ac.jp (J.N.); nozaki-n@juntendo.ac.jp (N.N.); yasu@mtnet.jp (Y.T.); yusuke@juntendo.ac.jp (Y.S.); 2Department of Nephrology, Juntendo University Nerima Hospital, Tokyo 177-8521, Japan; 3Department of Nephrology, Juntendo University Shizuoka Hospital, Shizuoka 410-2211, Japan

**Keywords:** l-carnitine, hemodialysis, peritoneal dialysis, renal anemia, muscle spasm

## Abstract

l-carnitine is an important factor in fatty acid metabolism, and carnitine deficiency is common in dialysis patients. This study evaluated whether l-carnitine supplementation improved muscle spasm, cardiac function, and renal anemia in dialysis patients. Eighty Japanese outpatients (62 hemodialysis (HD) patients and 18 peritoneal dialysis (PD) patients) received oral l-carnitine (600 mg/day) for 12 months; the HD patients further received intravenous l-carnitine injections (1000 mg three times/week) for 12 months, amounting to 24 months of treatment. Muscle spasm incidence was assessed using a questionnaire, and cardiac function was assessed using echocardiography. Baseline free carnitine concentrations were relatively low in patients who underwent dialysis for >4 years. Total carnitine serum concentration, free carnitine, and acylcarnitine significantly increased after oral l-carnitine treatment for 12 months, and after intravenous l-carnitine injection. There was no significant improvement in muscle spasms, although decreased muscle cramping after l-carnitine treatment was reported by 31% of patients who had undergone HD for >4 years. Hemoglobin concentrations increased significantly at 12 and 24 months in the HD group. Therefore, l-carnitine may be effective for reducing muscle cramping and improving hemoglobin levels in dialysis patients, especially those who have been undergoing dialysis for >4 years.

## 1. Introduction

Previous studies have indicated that dialysis patients are predisposed to developing carnitine deficiency associated with skeletal muscle weakness, cardiomyopathy, heart failure, and renal anemia. Furthermore, a randomized controlled trial revealed that carnitine treatment was associated with a significantly improved survival rate among dialysis patients with dilated cardiomyopathy [1]. These relationships may be because carnitine plays an important role in mitochondrial transport and long-chain fatty acid metabolism, contributing to energy production [2]. Dialysis patients may be predisposed to carnitine deficiency because of their protein-restricted diet, decreased l-carnitine production because of renal impairment, and increased carnitine loss during dialysis. Furthermore, dialysis patients typically have low total carnitine (TC) and free carnitine (FC) concentrations, while acylcarnitine (AC) is within the normal range. However, serum carnitine concentrations tend to decrease at longer dialysis duration [3]. In this setting, carnitine deficiency may reduce fatty acids, an energy source, leading to fat accumulation, muscle weakness, cardiomyopathy, heart failure, and anemia (related to erythrocyte membrane weakening) [4]. Carnitine is an amino acid-derived substance present in almost every cell in the body. The name carnitine is derived from the Latin word “carnus,” which means meat because it is a substance contained in meat. Carnitine is a general term for many substances such as l-carnitine, acetyl-l-carnitine, and propionyl-l-carnitine. Healthy children and adults do not need to take daily carnitine from food or supplements because the liver and kidneys synthesize sufficient amounts of the amino acid lysine and methionine [5,6]. The AC/FC ratio is an index used to evaluate carnitine metabolism. AC removal by dialysis is more difficult than FC removal, and patients with renal failure are believed to accumulate AC easily. As a result, even if we administered carnitine to improve the metabolism, the AC/FC level may not change appropriately. It has been reported in the past that the AC/FC level significantly decreased six months after switching to the IV dosing mode [7,8]. Therefore, the present study evaluated whether l-carnitine supplementation may improve muscle spasms, cardiac function, and renal anemia in Japanese dialysis patients.

## 2. Materials and Methods

This prospective study recruited 80 Japanese dialysis patients, including 62 patients who were undergoing hemodialysis (HD) and 18 patients who were undergoing peritoneal dialysis (PD) (Table 1). All patients were treated as outpatients at our hospital for dialysis and provided informed consent before participating in the study. The study protocol was approved by the Ethics Review Committee of Juntendo University Faculty of Medicine, Tokyo, Japan (approval: 12-014, 20 April 2013), and was registered with the University Hospital Medical Information Network (UMIN00007874, UMIN000012390). All study procedures complied with the tenets of the 2000 Declaration of Helsinki.

### Study Design

All 80 patients received oral l-carnitine (600 mg/day; Otsuka Pharmaceutical Co, Ltd. Tokyo, Japan) for 12 months, and then HD patients received l-carnitine injections (1000 mg three times per week) for 12 additional months. Data regarding patient clinical characteristics, serum samples, and the use of erythropoiesis-stimulating agents (ESAs) were obtained at the start of the study and then every 6 months. Cardiac parameters were evaluated using echocardiography and chest radiography at the beginning of the study and the final follow-up. A questionnaire was used to collect data regarding muscle spasms and cramps frequency at the start of the study and then every 6 months.

Blood testing was performed to determine the serum concentrations of TC, FC, and AC by enzyme cycling methods, as described previously [9], hemoglobin (Hb), hematocrit, iron, ferritin, C-reactive protein, brain natriuretic peptide, and atrial natriuretic peptide, and total iron-binding capacity. The echocardiography data were used to determine the left ventricular mass index, ejection fraction, and dilatation abilities of the left ventricle, which included the peak early diastolic left ventricular filling velocity/peak atrial filling velocity ratio, deceleration time, and peak early diastolic left ventricular filling velocity/peak early diastolic mitral annular velocity ratio.

## 3. Results

### 3.1. Effects of Oral l-Carnitine on Carnitine Concentrations

Normal serum concentrations were 45–91 µmol/L for TC, 36–74 µmol/L for FC, and 6–23 µmol/L for AC. At baseline, average serum concentrations were 45.8 ± 14.7 µmol/L for TC (HD: 45.4 ± 14.3 µmol/L, PD: 47.0 ± 16.1 µmol/L), 29.3 ± 9.9 µmol/L for FC (HD: 28.0 ± 10.0 µmol/L, PD: 34.0 ± 7.9 µmol/L), and 17.1 ± 5.3 µmol/L for AC (HD: 17.5 ± 5.5 µmol/L, PD: 15.7 ± 4.7 µmol/L).

Patients were categorized according to their dialysis duration due to their significantly different lengths (<1 year, 1–4 years, and >4 years). Baseline FC concentrations were low, especially in patients who had been undergoing dialysis for more than four years (Figure 1 and Figure 2a,b). TC and FC concentrations were also low, especially in patients who had been undergoing PD for more than 1.5 years (Figure 2c). After six months of oral l-carnitine treatment, average serum concentrations increased to 158.8 ± 74.0 µmol/L for TC (HD: 168.3 ± 75.0 µmol/L, PD: 125.2 ± 61.3 µmol/L), 100.8 ± 46.8 µmol/L for FC (HD: 105.8 ± 47.8 µmol/L, PD: 83.0 ± 39.8 µmol/L), and 58.0 ± 28.9 µmol/L for AC (HD: 62.4 ± 29.0 µmol/L, PD: 42.2 ± 23.2 µmol/L). Furthermore, after 12 months of oral treatment, serum concentrations increased to 170.5 ± 79.1 µmol/L for TC (HD: 182.1 ± 76.4 µmol/L, PD: 120.5 ± 73.6 µmol/L), 112.9 ± 51.1 µmol/L for FC (HD: 116.0 ± 52.3 µmol/L, PD: 98.5 ± 44.6 µmol/L), and 59.9 ± 27.7 µmol/L for AC (HD: 63.8 ± 27.6 µmol/L, PD: 40.9 ± 20.1 µmol/L). After 24 months of treatment for HD patients, concentrations increased to 255.7 ± 268.4 µmol/L for TC, 157.1 ± 163.3 µmol/L for FC, and 99.8 ± 106.7 µmol/L for AC (Figure 2d).

### 3.2. Effects of l-Carnitine on Muscle Symptoms

Responses to the questionnaire indicated that 58% of patients experienced muscle spasms, while 42% did not. Some patients reported feeling that their muscle spasms had been cured after 24 months of l-carnitine treatment, especially among those who had undergone dialysis for more than four years. We did not detect any significant improvement among patients who received oral l-carnitine for 12 months, although 31% of patients who had undergone dialysis for more than four years reported experiencing leg cramps improvements. This proportion was substantially larger than the proportion among patients who had undergone dialysis for four years or less (Figure 3). We categorized the patients into diabetes mellitus (DM) and non-DM groups and re-analyzed them. Regarding muscle spasms, non-DM patients were cured in 45.8%, unchanged in 43.8%, worsened in 6.3%, and unclear in 4.1% of cases; meanwhile, DM patients were cured in 7.7%, unchanged in 76.9%, worsened in 7.7%, and unclear in 7.7% of cases. Carnitine administration seemed to be effective in non-DM patients. There was no significant difference in the effect of carnitine administration between DM and non-DM patients for cardiac function and renal anemia.

### 3.3. Effects of Oral l-Carnitine on Cardiac Function

We did not observe any significant changes in echocardiography findings, although a small hemoglobin concentration increase in the HD group was observed (Table 2).

### 3.4. Effects of Oral l-Carnitine on Renal Anemia

At baseline, the average Hb concentration was 10.4 ± 1.1 g/dL (HD: 10.2 ± 1.2 g/dL, PD: 10.6 ± 1.1 g/dL). After six months of l-carnitine treatment, the average Hb concentration was 10.8 ± 0.8 g/dL (HD: 10.8 ± 0.8 g/dL, PD: 10.8 ± 0.9 g/dL). Average concentrations were 10.8 ± 1.0 g/dL after 12 months (HD: 10.9 ± 0.9 g/dL, PD: 10.6 ± 1.3 g/dL) and 10.7 ± 0.8 g/dL after 24 months (all HD cases) (Figure 4). The overall increase was significant at both 12 months (HD and PD cases) and 24 months (only HD cases).

We also considered ESA doses, with conversion to epoetin equivalents if necessary (darbepoetin alfa dose = 200× epoetin dose, epoetin beta pegol at 100 µg/week = 4500 units/week of epoetin, epoetin beta pegol at 150 µg/week = 6000 units/week of epoetin). At baseline, the average ESA dose was 6153.1 ± 5316.8 units/week (HD: 6080.6 ± 5866.8 units/week, PD: 6402.8 ± 2788.0 units/week). After six months of l-carnitine treatment, the average ESA dose was 6354.7 ± 8704.1 units/week (HD: 5343.8 ± 5442.2 units/week, PD: 10,058.8 ± 14,939.1 units/week) and after 12 months of treatment the average dose was 4880.6 ± 5649.4 units/week (HD: 4703.7 ± 5926.8 units/week, PD: 5615.4 ± 4434.3 units/week) (Figure 5). No significant change in the ESA dose was observed after oral or intravenous treatment using l-carnitine.

Finally, we considered the erythropoietin resistance index (ERI), which is calculated as (epoetin dose)/(bodyweight)/(hemoglobin concentration). At baseline, the average ERI was 10.9 ± 11.0 (HD: 11.1 ± 12.3, PD: 10.3 ± 5.0). After six months of l-carnitine treatment, the average ERI was 11.6 ± 14.8 (HD: 10.3 ± 11.3, PD: 15.922.6), and after 12 months, it was 9.5 ± 13.9 (HD: 9.4 ± 14.8, PD: 9.8 ± 10.0) (Figure 5). There was a slight but not significant decrease in the ERI values at 12 months and 24 months.

## 4. Discussion

Supplementation using l-carnitine has been widely reported as a potential option for managing various symptoms and physical function parameters in dialysis patients. A previous report has indicated that dialysis patients have reduced physical capacity, relative to comparable non-dialysis individuals [10], although that study only included a questionnaire regarding l-carnitine supplementation and muscle symptoms. Other reports have examined l-carnitine supplementation effects on muscle symptoms in dialysis patients, based on muscle histology, physical performance, muscle cramping incidences, and self-reported measures of functional capacity [11]. However, we did not observe any significant improvement in muscle spasms after 12 months of treatment using oral l-carnitine, although an improvement was reported by a subset of patients (31%) who had undergone dialysis for more than four years, and this proportion was substantially larger than the proportion among patients who had undergone dialysis for four years or less. We also observed relatively low baseline serum FC concentrations, especially among patients who had undergone dialysis for more than four years, as well as low baseline serum TC and FC concentrations among patients who had undergone PD for more than 1.5 years. Thus, relative carnitine deficiency may have been present at baseline in the PD group, which may hypothetically be related to carnitine loss during PD drainage. It is also possible that carnitine deficiency was related to dietary restrictions regarding protein and phosphorus. Given that carnitine deficiency was common for patients with a relatively long dialysis duration, carnitine supplementation may benefit those patients. Carnitine deficiency in dialysis patients may result in a loss of effluent. Carnitine is a water-soluble substance with a molecular weight of 162 g and is easily removed by permeation. It is expected that TC and FC will be accumulated in HD patients with reduced AC compared to healthy subjects. In PD patients, the AC concentration at the start was low, and it was more difficult to remove than TC and FC, so AC increased the most. It is considered that the low accumulation of AC in PD patients is due to the preservation of residual renal function. In the normal kidney, excess acyl groups were considered to be excreted as AC. A multicenter trial on 537 patients showed that propionyl-l-carnitine improves exercise capacity in patients with heart failure but preserved cardiac function [12].

Cardiovascular disease is a significant cause of death among dialysis patients [13]. A study in rats revealed that l-carnitine treatment significantly reduced skeletal myocyte apoptosis in rats with congestive heart failure [14]. A study of long-term dialysis patients revealed that l-carnitine supplementation (3 mg intravenously after each dialysis session) provided significant increases in the total output of surface electromyography activity (a measure of muscle function) after one and eight months, relative to the placebo treatment [15]. A previous study revealed that oral l-carnitine supplementation helped significantly increase the cardiac ejection fraction (EF) in patients with reduced EF [16]. Another study revealed that HD patients with symptomatic dialysis hypotension had relative carnitine deficiency and a reduced mean EF, relative to asymptomatic patients [17]. These findings may suggest that carnitine supplementation may help increase EF and/or prevent cardiac events in at-risk patients. In contrast, we failed to identify significant differences in echocardiography findings regarding cardiac function, although we did not perform an adjustment for the severity of cardiac dysfunction or coronary artery sclerosis.

Approximately 40% of dialysis patients fail to achieve a hemoglobin concentration of more than 11 g/dL, and up to 25% of patients receive ESA doses of 270 units/kg/week or more [18]. In this study, the Hb levels of 11.0 g/dL or higher at the start were 28.8%, and there was an improvement to 55.2% after 12 months for carnitine administration. Patients receiving an ESA dose of 270 unit/kg/week or more were 18.1% at the start and 19.6% after 12 months. Failure to achieve adequate hematocrit concentrations is also associated with poor survival and quality of life [19]. A placebo-controlled double-blind study evaluated the required ESA dose to maintain a hematocrit concentration of 28–33% in 24 patients (13 patients received 1 g intravenous l-carnitine per dialysis session for six months, 11 patients received placebo). The study revealed no change after six months in the placebo group and a significantly reduced ESA dose in the l-carnitine-treated group [20]. The present study revealed that HD patients experienced a significant increase in Hb concentration and a significant decrease in ESA dose. However, we failed to detect a significant decrease in the ERI among all patients or any significant changes among PD patients. Seventy-eight percent of PD patients had a history of PD for 18 months or longer, and the reason for the lack of improvement in Hb levels compared to HD patients may be due to low carnitine levels (shown in Figure 2c) in this study. We could not clarify why there was an improvement in Hb levels in HD patients but no significant improvement in PD patients. Carnitine administration significantly increased blood carnitine levels in both HD and PD patients. However, the rate of increase was lower in PD patients (although there was no significant difference) (Figure 6). In this regard, we considered it to disappear in the urine or PD drainage. Further studies are also needed to adjust for other factors associated with anemia, such as iron administration, gastrointestinal bleeding, inflammation, or malignancy.

The prospect that exposure to specific dietary nutrients like l-carnitine via gut microbiota may impact susceptibility to the development and progression of both CKD and CVD has important potential public health implications. Carnitine oxidoreductase is the main enzyme responsible for converting l-carnitine into trimethylamine-*N*-oxide (TMAO) [21,22]. Chronic oral l-carnitine supplementation was associated with increased trimethylamine-*N*-oxide (TMAO) levels through microbiota metabolism, and it might be harmful to the cardiovascular function in patients with kidney disease. This phenomenon is more pronounced in carnivorous people than in vegans and vegetarians. The implications of these findings are not fully understood, and further research is needed. Carnitine supplementation at doses of approximately 3 g/day may cause side effects such as nausea, vomiting, abdominal cramps, diarrhea, and “fishy” body odor [5,6].

Several limitations must be considered when interpreting our findings. First, the small-sample, uncontrolled, and non-randomized study design is prone to bias. Second, this study prospectively determined the duration of l-carnitine treatment. Third, patients might have had various types of renal disease. Fourth, muscle symptoms, cardiac function, and renal anemia in dialysis patients may be influenced by various medications, including statins, antiplatelet drugs, and angiotensin II receptor blockers.

## 5. Conclusions

The present study revealed that baseline carnitine concentrations were low among Japanese dialysis patients, especially among patients who had undergone dialysis for more than four years. However, l-carnitine oral and intravenous administrations were associated with significant increases in serum carnitine concentrations, especially among patients who had undergone dialysis for more than four years. Furthermore, l-carnitine treatment may help improve Hb concentrations and reduce muscle spasms in select patients.

## Figures and Tables

**Figure 1 nutrients-12-03371-f001:**
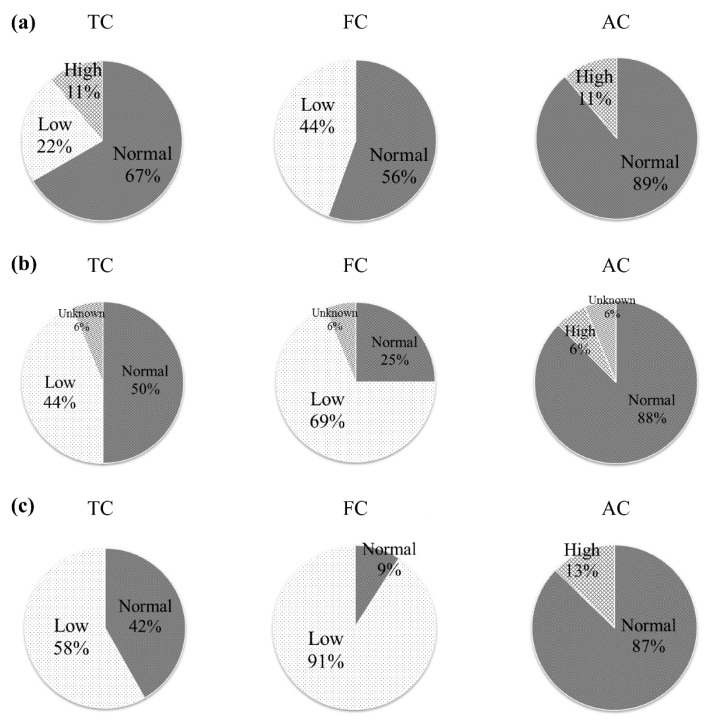
Baseline carnitine concentrations according to dialysis durations ranging from (**a**) <1 year, (**b**) 1–4 years, and (**c**) >4 years. Free carnitine (FC) concentrations were lower than total carnitine (TC) concentrations. AC: acyl carnitine. Normal serum concentrations were 45–91 µmol/L for TC, 36–74 µmol/L for FC, and 6–23 µmol/L for AC.

**Figure 2 nutrients-12-03371-f002:**
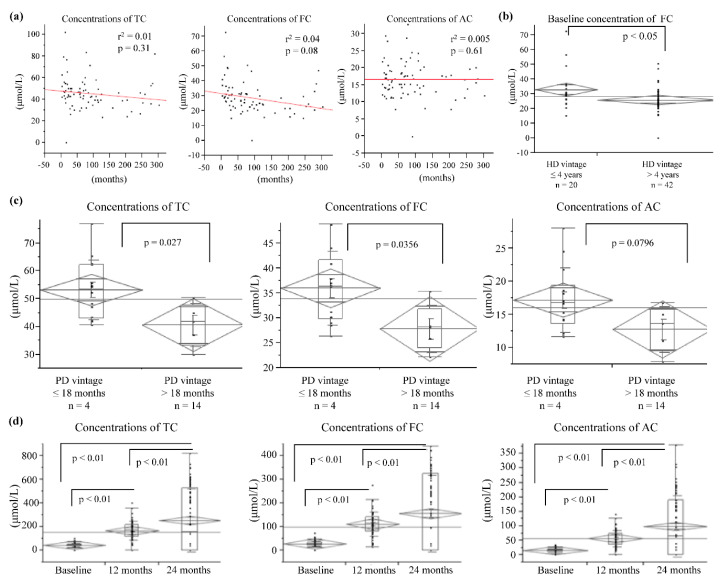
(**a**) Baseline concentrations of total carnitine (TC), free carnitine (FC), and acylcarnitine (AC) according to dialysis duration. (**b**) According to dialysis duration, FC concentrations at baseline revealed low baseline FC concentrations for patients with a hemodialysis (HD) duration of more than 4 years. (**c**) Baseline carnitine levels concentrations, according to peritoneal dialysis (PD) duration. (**d**) Changes in carnitine concentrations during l-carnitine treatment, which revealed significant TC, FC, and AC increases.

**Figure 3 nutrients-12-03371-f003:**
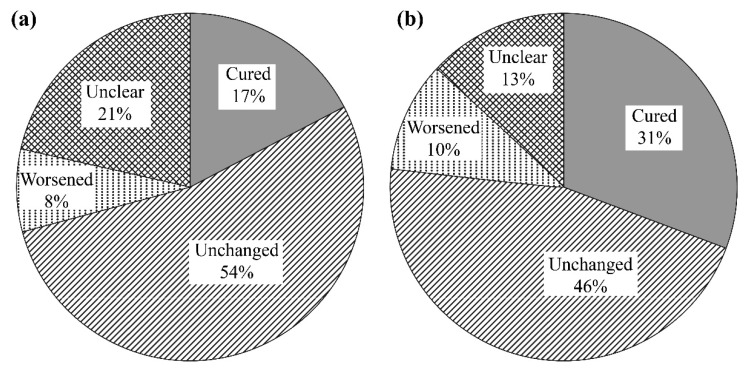
Survey responses regarding changes in muscle spasms according to dialysis durations of (**a**) ≤4 years and (**b**) >4 years. The proportion of patients who reported that leg cramps had been cured was substantially larger among patients who had undergone dialysis for >year.

**Figure 4 nutrients-12-03371-f004:**
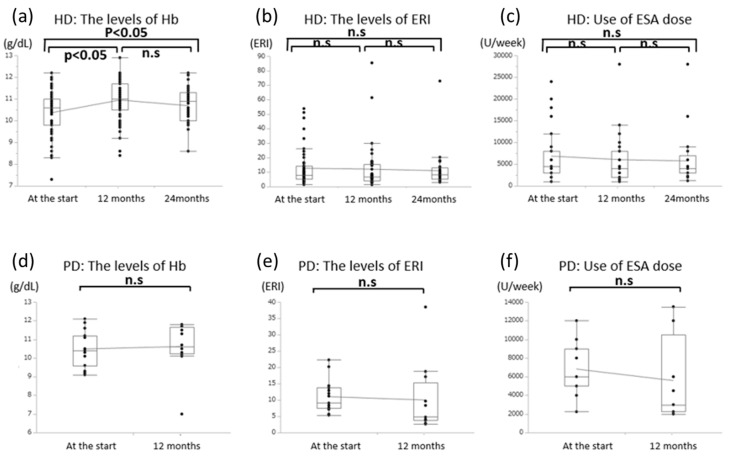
(**a**) Effects of l-carnitine treatment on hemoglobin concentrations, which significantly increased after 12 months and 24 months in HD patients. (**b**) The levels of ERI in HD patients. (**c**) Use of ESA dose in HD patients. (**d**) The levels of Hb in PD patients. (**e**) The levels of ERI in PD patiemts. (**f**) Use of ESA dose in PD patients.

**Figure 5 nutrients-12-03371-f005:**
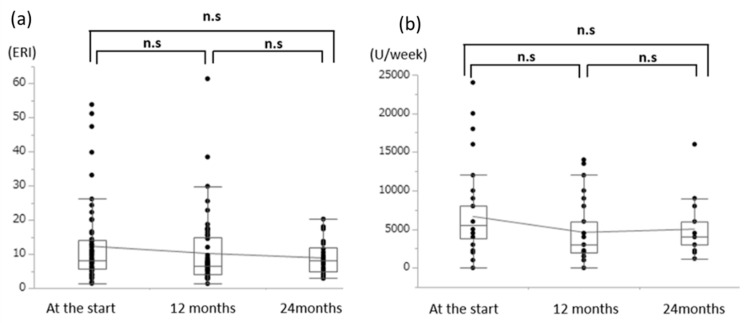
Treatment using l-carnitine did not significantly reduce the average erythropoietin resistance index (**a**) or erythropoiesis-stimulating agents (ESA) dose (**b**). ESA: erythropoiesis-stimulating agents, ERI: erythropoietin resistance index ([epoetin dose]/[body weight]/[hemoglobin concentration]).

**Figure 6 nutrients-12-03371-f006:**
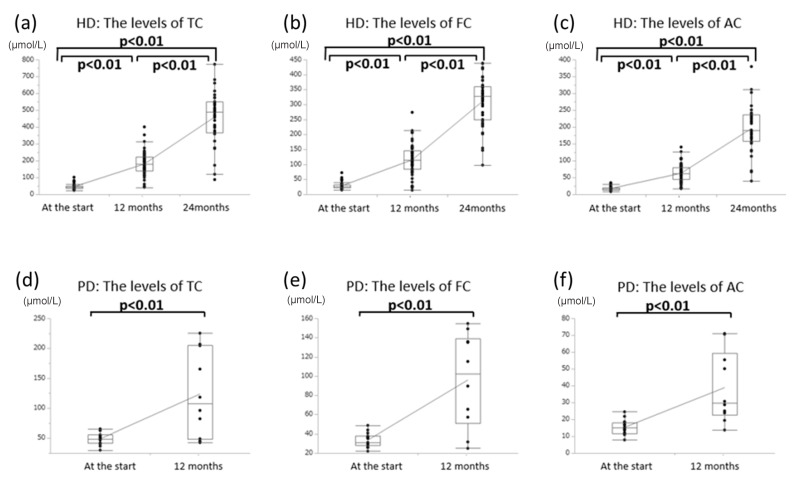
Carnitine administration significantly increased blood carnitine levels in both HD (**a**–**c**) and PD patients (**d**–**f**).

**Table 1 nutrients-12-03371-t001:** Baseline characteristics of the 80 patients.

	All Patients (*n* = 80)	PD (*n* = 18)	HD (*n* = 62)
Age (years)	62.4 ± 14.8	56.7 ± 11.7	64.0 ± 15.3
Dialysis duration (months)	88.1 ± 85.4	14.8 ± 11.1	112.5 ± 85.4
Diabetes mellitus	18/80	0/18	18/62
Male sex (%)	78.8	66.1	66.7

HD: hemodialysis, PD: peritoneal dialysis.

**Table 2 nutrients-12-03371-t002:** Cardiac-related parameters.

	Peritoneal Dialysis		Hemodialysis		
	0 Months	12 Months	0 Months	12 Months	24 Months
Hb (g/dL)	10.6 ± 1.1	10.6 ± 1.3	10.2 ± 1.2 *	10.9 ± 0.9 *	10.7 ± 0.8 *
LVMI ^1^ (g/m ^2^)	142.4 ± 46.8	133.6 ± 37.4	152.5 ± 37.8	151.2 ± 36.6	153.3 ± 36.8
E/A ^2^	0.83 ± 0.35	0.70 ± 0.37	0.77 ± 0.28	0.75 ± 0.20	0.75 ± 0.19
E/e’^3^	10.0 ± 3.6	10.4 ± 5.0	13.9 ± 5.5	15.1 ± 7.2	16.2 ± 7.3
BNP ^4^ (pg/mL)	121.5 ± 119	130.7 ± 140	455.0 ± 40.3	446.8 ± 57.8	449 ± 56.7
ANP ^5^ (pg/mL)	54.3 ± 41.45	52.3 ± 38.3	84.6 ± 66.5	74.8 ± 58.4	79.2 ± 59.3
LDL ^6^ (mg/dL)	95.0 ± 22.3	92.0 ± 21.4	83.0 ± 28.4	89.0 ± 24.5	89.7 ± 24.8
HDL ^7^ (mg/dL)	48.0 ± 10.2	58.0 ± 18.1	44.0 ± 12.2	45.0 ± 11.5	44.9 ± 12.8
TG ^8^ (mg/dL)	104.0 ± 33.6	101.0 ± 36.7	101.0 ± 39.8	122.0 ± 55.8	125.6 ± 56.7
TSAT ^9^ (%)	35.8 ± 8.8	27.6 ± 11.4	21.8 ± 11.5	24.4 ± 12.1	25.6 ± 12.7

^1^ left ventricular mass index; ^2^ peak early diastolic left ventricular filling velocity/peak atrial filling velocity ratio; ^3^ peak early diastolic left ventricular filling velocity/peak early diastolic mitral annular velocity ratio; ^4^ brain natriuretic peptide; ^5^ atrial natriuretic peptide; ^6^ low-density lipoprotein; ^7^ high-density lipoprotein; ^8^ triglyceride; ^9^ transferrin saturation.

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
