# Peer review of "Effects of l-Carnitine Supplementation in Patients Receiving Hemodialysis or Peritoneal Dialysis"

_nutrients, 2020, doi:10.3390/nu12113371_

Round 1
Reviewer 1 Report
This study investigated that the effects of L-carnitine supplementation in patients receiving hemodialysis or peritoneal dialysis. The authors provided evidence that L-carnitine may be effective for reducing muscle cramping and improving hemoglobin levels in dialysis patients.
Major points:
- In the study design the authors mentioned that blood testing was performed to determine the serum concentration of TC, FC, and AC. Can you give us more detailed what kind of instrument or method analysis was used to quantify these carnitines? For example, HPLC, LC-MS or immunoassay is widely used to quantify L-carnitine. Method analysis is very important to confirm that the analytical procedure employed for a specific test or quantification is suitable for its intended use.
- In Table 1, there are 18 diabetes patients were in the HD group. Why the authors included these patients since there were enough patients in the HD group (44 patients without diabetes). This reviewer was wondering if there any significant difference in carnitine level (particularly), muscle spasm, cardiac function, and renal anemia in dialysis patients after L-carnitine treatment if these 18 DM patients were excluded?
- Add more information in the introduction part and increasing other related references.
Minor points:
- Is there any particular reason why the authors also quantified acylcarnitine (AC)? The author did not mention the difference, importance or significance of these different forms of carnitine. How about other carnitine derivatives? Such as “propiony-L-carnitine”.
- There was inconsistent data of Hb value between Table 2 and results in section 3.4 (effects of oral L-carnitine on renal anemia). In Table 2, the concentration of Hb after 12 months were 9±0.9 for HD, 10.6±1.3 for PD, and 10.7±0.8 after 24 months in HD patients. But in section 3.4, after 12 months, the concentration of Hb after 12 months was 11.0±0.9 g/dL for HD, 10.6±1.2 g/dL for PD and 10.7±0.7 g/dL after 24 months (all HD cases).
- In previous many studies demonstrated that chronic oral L-carnitine supplementation was associated with increased trimethylamine-N-oxide (TMAO) levels through microbiota metabolism, it might be harmful to the cardiovascular function in patients with kidney disease. This reviewer suggested that the authors should have more discussions for this.
Author Response
Dear Sir,
Re: nutrients-967389 I am enclosing for possible evaluation an original article entitled “Effects of L-carnitine supplementation in patients receiving hemodialysis or peritoneal dialysis” for publication in Nutrients.
Thank you for your letter of October 7, 2020 and for the reviewer’s comments concerning the above manuscript. We would like to thank the reviewers and the editor for the thorough revision of our paper, which we indeed believe has improved it.
The revised parts are underlined. Responses to the Editor and Reviewers and an index of the changes are included on separate pages.
Your consideration in this matter will be appreciated.
Sincerely Yours,
Hiroaki Io
The paper was revised according to the suggestions made. According to the reviewer’s comments, we changed our manuscript. All changes were made in color and underline in the revised manuscript. Some new comments and changes in the text were added. We sincerely believe that we have adequately answered all queries and issues. We would like to thank the Editor and Reviewers for the thorough revision of our paper, which we indeed believe has improved it.
Reviewer’s 1 comments and point by point answers
Major points:
- In the study design the authors mentioned that blood testing was performed to determine the serum concentration of TC, FC, and AC. Can you give us more detailed what kind of instrument or method analysis was used to quantify these carnitines? For example, HPLC, LC-MS or immunoassay is widely used to quantify L-carnitine. Method analysis is very important to confirm that the analytical procedure employed for a specific test or quantification is suitable for its intended use.
<Answer>
Serum carnitine levels were determined by enzyme cycling methods, as described previously (ref).
ref: Takahashi M, Ueda S, Misaki H, Sugiyama N, Matsumoto K, Matsuo N, Murao S.: Carnitine determination by an enzymatic cycling method with carnitine dehydrogenase. Clin Chem 1994;40:817–821
This sentence was added in “Method “(2.1 Study design) and reference. (Lines 68-69)
- In Table 1, there are 18 diabetes patients were in the HD group. Why the authors included these patients since there were enough patients in the HD group (44 patients without diabetes). This reviewer was wondering if there any significant difference in carnitine level (particularly), muscle spasm, cardiac function, and renal anemia in dialysis patients after L-carnitine treatment if these 18 DM patients were excluded?
<Answer>
As you pointed out, we categorized the patients into diabetes mellitus (DM) and non-DM groups and re-analyzed them. Regarding muscle spasms, non-DM patients were cured in 45.8%, unchanged in 43.8%, worsened in 6.3%, and unclear in 4.1% of cases, respectively, meanwhile, DM patients were cured in 7.7%, unchanged in 76.9%, worsened in 7.7%, and unclear in 7.7% of cases, respectively. Carnitine administration seemed to be effective in non-DM patients. There was no significant difference in the effect of carnitine administration between DM and non-DM patients for cardiac function and renal anemia.
These sentences were added to “Results.” (Lines 112-117)
- Add more information in the introduction part and increasing other related references.
<Answer>
Carnitine is an amino acid-derived substance present in almost every cell in the body. The name carnitine is derived from the Latin word "carnus," which means meat because it is a substance contained in meat. Carnitine is a general term for many substances such as L-carnitine, acetyl-L-carnitine, and propionyl-L-carnitine. Healthy children and adults do not need to take daily carnitine from food or supplements because the liver and kidneys synthesize sufficient amounts of the amino acid lysine and methionine. (ref1,2). The AC/FC ratio is an index used to evaluate carnitine metabolism. AC removal by dialysis is more difficult than FC removal, and patients with renal failure are believed to accumulate AC easily. As a result, even if we administered carnitine to improve the metabolism, the AC/FC level may not change appropriately. It has been reported in the past that the AC/FC level significantly decreased 6 months after switching to the IV dosing mode (ref3,4).
(ref1) Rebouche CJ. Carnitine. In: Modern Nutrition in Health and Disease, 9th Edition (edited by Shils ME, Olson JA, Shike M, Ross, AC). Lippincott Williams and Wilkins, New York, 1999, pp. 505-12.
(ref2) The editors. Carnitine: lessons from one hundred years of research. Ann NY Acad Sci 2004;1033:ix-xi.
(ref3) Fukami K, Sakai K, Kaida Y, et al. . Effects of oral or intravenous L-carnitine administration on serum carnitine levels and clinical parameters in hemodialysis patients. J Jpn Soc Dial Ther. 2014;47:367–374.
(ref4) Fukami K, Yamagishi S, Sakai K, et al. . Effects of switching from oral administration to intravenous injection of L-carnitine on lipid metabolism in hemodialysis patients. Clin Kidney J. 2014;7:470–474.
These sentences were added to “Introduction” and reference. (Lines 38-47)
Minor points:
- Is there any particular reason why the authors also quantified acylcarnitine (AC)? The author did not mention the difference, importance or significance of these different forms of carnitine. How about other carnitine derivatives? Such as “propiony-L-carnitine”.
<Answer>
Carnitine deficiency in dialysis patients may result in a loss of effluent. Carnitine is a water-soluble substance with a molecular weight of 162 and is easily removed by permeation. It is expected that TC and FC will be accumulated in HD patients with reduced AC compared to healthy subjects. In PD patients, the AC concentration at the start was low, and it was more difficult to remove than TC and FC, so AC increased the most. It is considered that the low accumulation of AC in PD patients is due to the preservation of residual renal function. In the normal kidney, excess acyl groups were considered to be excreted as AC. A multicenter trial on 537 patients showed that propionyl-L-carnitine improves exercise capacity in patients with heart failure but preserved cardiac function (ref).
(ref) Ferrari R, Merli E, Cicchitelli G, Mele D, Fucili A, Ceconi C.: Therapeutic effects of L-carnitine and propionyl-L-carnitine on cardiovascular diseases: a review. Ann N Y Acad Sci. 2004 Nov; 1033:79-91
These sentences were added to “Discussion” and reference. (Lines 182-190)
- There was inconsistent data of Hb value between Table 2 and results in section 3.4 (effects of oral L-carnitine on renal anemia). In Table 2, the concentration of Hb after 12 months were 9±0.9 for HD, 10.6±1.3 for PD, and 10.7±0.8 after 24 months in HD patients. But in section 3.4, after 12 months, the concentration of Hb after 12 months was 11.0±0.9 g/dL for HD, 10.6±1.2 g/dL for PD and 10.7±0.7 g/dL after 24 months (all HD cases).
<Answer>
It is as you pointed out. I made a mistake in the data on the way and wrote it in the text. The final data is definitely in Table 2. I rewrote the sentences as follows.
“At baseline, the average Hb concentration was 10.4±1.1 g/dL (HD: 10.2±1.2 g/dL, PD: 10.6±1.1 g/dL). After 6 months of L-carnitine treatment, the average Hb concentration was 10.8±0.8 g/dL (HD: 10.8±0.8 g/dL, PD: 10.8±0.9 g/dL). Average concentrations were 10.8±1.0 g/dL after 12 months (HD: 10.9±0.9 g/dL, PD: 10.6±1.3 g/dL) and 10.7±0.8 g/dL after 24 months (all HD cases) (Figure 4). (Lines 132-135)
I revised Fig 4 to clarify about HD and PD.
- In previous many studies demonstrated that chronic oral L-carnitine supplementation was associated with increased trimethylamine-N-oxide (TMAO) levels through microbiota metabolism, it might be harmful to the cardiovascular function in patients with kidney disease. This reviewer suggested that the authors should have more discussions for this.
<Answer>
The prospects that exposure to specific dietary nutrients like L-carnitine via gut microbiota may impact susceptibility to the development and progression of both CKD and CVD has important potential public health implications. Carnitine oxidoreductase is the main enzyme responsible for converting L-carnitine into trimethylamine-N-oxide (TMAO) (ref1,2). Chronic oral L-carnitine supplementation was associated with increased trimethylamine-N-oxide (TMAO) levels through microbiota metabolism, and it might be harmful to the cardiovascular function in patients with kidney disease. This phenomenon is more pronounced in carnivorous people than in vegans and vegetarians. The implications of these findings are not fully understood, and further research is needed.
(ref1) Zhu Y., Jameson E., Crosatti M., Schäfer H., Rajakumar K., Bugg T.D., Chen Y. Carnitine metabolism to trimethylamine by an unusual Rieske-type oxygenase from human microbiota. Proc. Natl. Acad. Sci. USA. 2014;111:4268–4273.
(ref2) Koeth RA, Wang Z, Levison BS, Buffa JA, Org E, Sheehy BT, Britt EB, Fu X, Wu Y, Li L, Smith JD, Didonato JA, Chen J, Li H, Wu GD, Lewis JD, Warrier M, Brown JM, Krauss RM, Tang WH, Bushman FD, Lusis AJ, Hazen SL. Intestinal microbiota metabolism of L-carnitine, a nutrient in red meat, promotes atherosclerosis. Nat Med. 2013 May;19(5):576-85.
These sentences were added to “Discussion” and reference. (Lines 222-229)

Reviewer 2 Report
This paper describes this study of carnitine supplementation in dialysis patients clearly and effectively.
The study design, as noted in the discussion, is not randomized, and (as should be noted) is uncontrolled. This is a major weakness in study design. It would add some strength if historical comparisons could be used, to show, for instance, that ESA requirements do not decrease with time in dialysis patients.
Specific comments:
As stated above, I would add that the"small-sample non-randomized study" is also uncontrolled.
I do not find the suggestion compelling that "lower serum carnitine baseline concentrations among PD patients might have influenced the Hb concentrations after carnitine treatment," since the concentrations were generally higher than in HD patients.
In the discussion, reference is made to the "40% of dialysis patients [who] fail to achieve a hemoglobin concentration of more than 11," and "up to 25% of patients [who] receive ESA doses of 270 units/kg/week or more," but in the results analysis, there is no description of the fraction of the patients in the study who failed to achieve these benchmarks. This would be a good analysis to include.
A comment about the significance of the supranormal levels of carnitine achieved with treatment would be appropriate.
Author Response
Dear Sir,
Re: nutrients-967389 I am enclosing for possible evaluation an original article entitled “Effects of L-carnitine supplementation in patients receiving hemodialysis or peritoneal dialysis” for publication in Nutrients.
Thank you for your letter of October 7, 2020 and for the reviewer’s comments concerning the above manuscript. We would like to thank the reviewers and the editor for the thorough revision of our paper, which we indeed believe has improved it.
The revised parts are underlined. Responses to the Editor and Reviewers and an index of the changes are included on separate pages.
Your consideration in this matter will be appreciated.
Sincerely Yours,
Hiroaki Io
The paper was revised according to the suggestions made. According to the reviewer’s comments, we changed our manuscript. All changes were made in color and underline in the revised manuscript. Some new comments and changes in the text were added. We sincerely believe that we have adequately answered all queries and issues. We would like to thank the Editor and Reviewers for the thorough revision of our paper, which we indeed believe has improved it.
Reviewer’s 2 comments and point by point answers
- The study design, as noted in the discussion, is not randomized, and (as should be noted) is uncontrolled. This is a major weakness in study design. It would add some strength if historical comparisons could be used, to show, for instance, that ESA requirements do not decrease with time in dialysis patients.
<Answer>
This study is not randomized, uncontrolled, and small-sample. This sentence has been added to “Discussion” (limitation). (Lines 232-233)
Although historical control data are inadequate, in the target group for which data could be aggregated, the ESA dose was about 5,000 units for HD patients and about 10,000 units for PD patients a year ago. There was no significant difference in HD patients, but in PD patients, it was low at n = 10 but tended to be low after LC administration.
- As stated above, I would add that the"small-sample non-randomized study" is also uncontrolled.
<Answer>
This study is not randomized, uncontrolled, and small-sample. This sentence is added to Discussion (limitation). (Lines 232-233)
- I do not find the suggestion compelling that "lower serum carnitine baseline concentrations among PD patients might have influenced the Hb concentrations after carnitine treatment," since the concentrations were generally higher than in HD patients.
<Answer>
As you pointed out, this sentence has been deleted because it does not show the data to prove.
Seventy eight percent of PD patients had a history of PD for 18 months or longer, and the reason for the lack of improvement in Hb levels compared to HD patients may be due to low carnitine levels (shown Fig2c) in this study. We could not clarify why there was an improvement in Hb levels in HD patients but no significant improvement in PD patients. Carnitine administration significantly increased blood carnitine levels in both HD and PD patients. However, the rate of increase was lower in PD patients (although there was no significant difference) (Fig.6). In this regard, we considered it to disappear in the urine or PD drainage.
These sentences were added to Discussion and added new Fig.6 (Lines 214-219)
- In the discussion, reference is made to the "40% of dialysis patients [who] fail to achieve a hemoglobin concentration of more than 11," and "up to 25% of patients [who] receive ESA doses of 270 units/kg/week or more," but in the results analysis, there is no description of the fraction of the patients in the study who failed to achieve these benchmarks. This would be a good analysis to include.
<Answer>
In this study, the Hb levels of 11.0 g / dl or higher at the start was 28.8%, and there was an improvement to 55.2% after 12 months for carnitine administration. Patients receiving ESA doss of 270 unit/kg /week or more were 18.1% at the start and 19.6% after 12 months.
These sentences were added to Discussion. (Lines 204-207)
- A comment about the significance of the supranormal levels of carnitine achieved with treatment would be appropriate.
<Answer>
Carnitine supplementation at doses of approximately 3 g / day may cause side effects such as nausea, vomiting, abdominal cramps, diarrhea, and "fishy" body odor (ref1,2 : Same as ref1 and 2 of the answer of reviewer 1-3.).
This sentence was added to Discussion. (Lines 204-207)

Round 2
Reviewer 1 Report
The author has answered all questions.